

# A study on elite athletes: Orthorexia nervosa tendency is a risk factor for sleep quality

Musab Çağın[1], Sezen Çimen Polat[2], Halil Sarol[3], Çisem Ünlü[4] and Danica Janicijevic[5,6,7]

[1] Department of Physical Education and Sports Teaching, Gazi University, Sport Sciences Faculty, Ankara, Turkey, Turkey

[2] Department of Coaching Education, Gazi University, Sport Sciences Faculty, Ankara, Turkey

[3] Department of Recreation, Gazi University, Sport Sciences Faculty, Ankara, Turkey

[4] Sport Management, Hitit University, Sport Science Faculty, Corum, Turkey

[5] Faculty of Sports Science, Ningbo University, Ningbo, China

[6] Department of Radiology, Ningbo No. 2 Hospital, Ningbo, China

[7] Department of Sports Sciences and Physical Conditioning, Faculty of Education, Universidad Catolica de la Santísima Concepción, Concepción, Chile

Corresponding author
Danica Janicijevic,
jan.danica@gmail.com

## ABSTRACT

**Background**. Elite athletes adhere to strict daily routines, particularly concerning their nutritional habits. However, these practices can occasionally lead to a pathological fixation on healthy eating known as orthorexia nervosa which may adversely affect sleep quality. The aim of this study was to examine the relationship between orthorexia nervosa tendencies and sleep quality in elite athletes, as well as to investigate potential differences in orthorexia nervosa tendencies and sleep patterns between athletes engaged in individual versus team sports.

**Methods**. The present study explores how orthorexia nervosa tendency affects sleep quality in 249 elite athletes (115 women: aged $19.92 \pm 2.48$ years, sporting tenure: $8.16 \pm 3.96$ years, and 134 men: aged $20.69 \pm 2.72$ years, sporting tenure: $7.85 \pm 3.59$ years). The orthorexia nervosa tendency was evaluated using the ORTO-11 scale, while their sleep quality using the Pittsburgh Sleep Quality Index.

**Results**. The results of the present study indicate that orthorexia nervosa negatively affected sleep quality (rho $= -0.173$, $p = 0.006$). Additionally, no differences in the orthorexia nervosa tendency was observed between individual and group athletes ($p = 0.287$); however, individual athletes presented poorer sleep quality ($p = 0.287$).

**Conclusion**. These results indicate that the elite athletes who present higher orthorexia nervosa tendency had a higher risk of having poorer sleep quality.

## INTRODUCTION

Sleep controls cognitive functions, neural activities, arousal, behavior and movement and can affect the biological system in numerous ways (*Çağın, 2021*). When sleep is examined in terms of sports, it can be expressed as the most crucial phase of the rest and recovery process after physical activity. It is particularly important for an athlete to be

physiologically, psychologically and cognitively rested and ready to perform at the desired level (*Çağın & Yarım, 2022*). Various studies examining the relationship between sleep and athletic performance have found that there is a positive relationship between the quality and duration of sleep and successful athletic performance (*Brandt, Bevilacqua & Andrade, 2017*; *Juliff et al., 2018*; *Andrade et al., 2021*). In addition to its enhancing effects on sportive performance, sleep also has a protective effect against diseases and athletic injuries (*Von Rosen et al., 2017*). Additionally, it is observed that athletes who sleep less than 8 h a day are 70% more likely to report an injury compared to athletes who sleep more than 8 h a day (*Milewski et al., 2014*). Considering the factors affecting sleep quality, it is observed that the factors related to the nutrition of the athlete stand out (*Helvacı& Ayhan, 2019*).

Eating disorders are commonly associated with poor sleep quality. A recent systematic review found that individuals with eating disorders experience significantly worse sleep quality compared to their healthy peers (*Degasperi et al., 2024*). *Kim et al. (2010)* reported that approximately 57% of individuals with eating disorders suffer from various sleep disturbances, including difficulty falling asleep, early morning awakenings, and mid-sleep awakenings. The link between eating disorders and poor sleep quality is further supported by clinical evidence suggesting that successful treatment of eating disorders can decrease sleep problems (*Lundgren et al., 2008*), and that addressing sleep issues while treating eating disorders may enhance the overall treatment trajectory (*Cooper, Loeb & McGlinchey, 2020*). Earlier research on the relationship between eating disorders and sleep quality predominantly focused on conditions such as anorexia and bulimia. However, a newly described eating disorder, orthorexia nervosa, has garnered increasing attention from researchers due to its relatively high prevalence in modern exercise-oriented populations (*Lauer & Krieg, 2004*). While no studies have yet established a direct link between orthorexia nervosa and sleep quality, its classification within the broader category of eating disorders suggests that individuals with this condition may also be prone to sleep-related disturbances.

A possible mechanism for suffering orthorexia nervosa, which is defined as an obsession with healthy nutrition, can be provoked by a great deal of stress regarding compliance with nutrition programs in the pre-competition period. Individuals with orthorexia nervosa may impose restrictions on the foods they consume or completely eliminate them from their diet on the grounds that they are unnatural or unhealthy (*Donini et al., 2004*). Such individuals tend to focus excessively on the technique and ingredients used in the preparation of food (*Moroze et al., 2015*). That is why it is not surprising that generally orthorexia nervosa and other eating disorders are more prevalent in elite athletes compared to the general population (*Haase, 2009*; *Martinsen & Sundgot-Borgen, 2013*; *Segura-García et al., 2012*; *Sundgot-Borgen & Torstveit, 2004*). Furthermore, the prevalence of eating disorders tends to be higher in athletes participating in individual sports compared to those in team sports (*Pamuk et al., 2020*). This can be attributed to the greater emphasis on individual physique, which is often subject to intense social scrutiny in individual sports (*Haase, 2009*). Additionally, performance in many individual sports is more directly linked to body weight and appearance (*e.g.*, aesthetic sports such as gymnastics, figure skating, and diving, or weight-sensitive sports like martial arts and ski jumping). In contrast, team sports such as volleyball and football place a stronger emphasis on tactics and teamwork, where body

weight is typically less central to performance. This difference in focus may explain the heightened vulnerability of athletes in individual sports to developing eating disorders.

Therefore, the aim of the study is (1) to explore relationship between orthorexia nervosa tendency and sleep quality in elite athletes, and (2) to explore differences between orthorexia nervosa tendency and sleep patterns between athletes of different branches (athletes participating in group *vs.* individual sports branches). We hypothesized that high orthorexia nervosa tendency will negatively affect sleep quality and that orthorexia nervosa would have a higher prevalence among athletes in individual sports compared to those in team sports.

## MATERIALS & METHODS

### Participants

A total of 249 elite athletes (115 women: age $= 19.92 \pm 2.48$ years, sporting tenure: $8.16 \pm 3.96$ years and 134 men: age $= 20.69 \pm 2.72$ years, sporting tenure: $7.85 \pm 3.59$ years). The power analysis technique was employed to ascertain the sample size for the study. Based on a two-tailed independent samples $t$-test analysis with a confidence level of 95% $(1-\alpha)$, a test power of 95% $(1-\beta)$, and an effect size of $d = 0.5$, the calculated number of samples for each group was determined to be 105. The participants were competing in several sport branches such as football, basketball, volleyball, handball, tennis, wrestling, taekwondo and swimming. Being an active elite athlete, having no history of injury in the last six months and not using any sleeping medication were the principal inclusion criteria The necessary permission and approval for the study was obtained from the Gazi University Ethics Commission (Code: 2023-1237) and the study was conducted in accordance with the Declaration of Helsinki. Participation in the study was voluntary. The participants signed a voluntary consent form and were informed about all details of the study.

### Procedures

Data were collected from participants in a face-to-face environment. Since the participants were athletes from different branches, a calendar was created by assigning appointments to different days for each branch. The participants were first given information about the research, signed voluntary participation forms, and were asked to answer the questions posed to them. The orthorexia nervosa tendencies of the subjects were detected using the ORTO-11 scale (*Bratman & Knight, 2000*), while their sleep quality was determined using the 24-item Pittsburgh Sleep Quality Index (*Ağargün, Kara & Anlar, 1996*). The detailed descriptions of the scales are provided in continuation:

*ORTO 11 Scale:* The ORTO-11 scale was used to determine the orthorexia nervosa tendencies of the athletes, which represents the Turkish adaptation of the originally developed 10 item scale developed by *Bratman & Knight (2000)*. ORTO-15 scale was adapted into Turkish by *Arusoğlu et al. (2008)* and was redesigned to a scale of 11 items. The 11-item scale is scored on a 4-point Likert scale, where lower scores indicate a higher tendency towards orthorexia. Individuals are asked to respond using the expressions "always (1), often (2), sometimes (3) and never (4)" to reflect how often they identify themselves with the statements given. The internal consistency of the ORTO-11 scale was

higher than that of the ORTO-15 scale, with Cronbach's alpha values of 0.62 and 0.44, respectively (*Arusoğlu et al., 2008*).

*Pittsburgh Sleep Quality Index (PSQI):* The Pittsburgh Sleep Quality Index (PSQI) was used to determine the sleep quality of the athletes. The first 19 of the 24 questions in the PSQI are self-report questions and the remaining five questions are answered by a roommate or spouse. The last five questions, which are exclusively used to obtain information in clinical studies, are not included in the scoring. The self-report questions include seven factors and are related to subjective sleep quality, sleep latency, sleep duration, habitual sleep efficiency, sleep disturbance, sleep medication use and daytime dysfunction, whose sum represents the total PDQI score (possible PSQI score is from 0 to 21, higher scores presented poorer sleep quality). PDQI total scores greater than 5 indicate poor sleep quality and scores less than 5 indicate good sleep quality. The PDQI is a quantitative measure of sleep quality used for the identification of "good sleep" and "poor sleep". The reliability and validity studies of the scale for Turkey were conducted by *Ağargün, Kara & Anlar (1996)*.

### Statistical analyses

The descriptive data regarding age and sporting tenure are presented using mean values and standard deviation, while the descriptive data related to ORTO-11 and PSQI scale are presented using median and interquartile range. Spearman correlation analysis was used for exploring the association between orthorexia nervosa tendency and sleep quality, as well between sports tenure and dependent variables (*i.e.,* orthorexia nervosa tendency and sleep quality). Furthermore, bivariant regression analysis was implemented to examine the predictive power of orthorexia nervosa tendency on sleep quality. Finally, Mann–Whitney U-test was used for exploring differences in orthorexia nervosa tendency and sleep quality between athletes practicing individual and team sports. Qualitative interpretations of the $r$ coefficients are defined as trivial (0.00–0.09), small (0.10–0.29), moderate (0.50–0.69), large (0.50–0.69), very large (0.70–0.89), nearly perfect (0.90–0.99), and perfect (1.00) (*Hopkins et al., 2009*). All statistical analysis is performed using SPSS 26.0 (IBM Corp., Armonk, NY). Alpha was set to $p \leq 0.05$.

## RESULTS

There is significant negative association between orthorexia nervosa tendency and sleep quality ($r_{ho} = -0.173$, $p = 0.006$). Furthermore, orthorexia nervosa tendency had significant negative predictive power on sleep quality ($\beta = -.127$; $t = -2.009$; p = .046) explaining 2% of the mutual common variance (Table 1, Fig. 1).

When comparing athletes participating in individual and in group sports results indicated that individual athletes had poorer sleep quality ($U = 5636.500$, $p \leq 0.001$), while orthorexia tendency was equalized between groups ($U = 7045.000$, $p = 0.287$) (Table 2).

## DISCUSSION

The principal aim of the present study was to explore the association between orthorexia nervosa tendency and sleep quality in elite individual and team sport athletes. The main

**Table 1  Regression analysis results for predicting sleep quality based on the orthorexia nervosa tendency.** This table shows the results of the bivariate regression that indicates that orthorexia nervosa tendency significantly affects sleep quality patterns in elite athletes.

| Model | B | Std. Error | β | t | p |
|---|---|---|---|---|---|
| Constant | 7.338 | .891 | — | 8.239 | $\leq$.001 |
| Orthorexia nervosa tendency | −.065 | .032 | −.127 | −2.009 | .046 |

Notes.

Dependent variable = Sleep quality; $R = 0.13$; $R^2_{adj} = 0.02$; $F_{(1,247)} = 4.035$; $p = 0.046$; Method: Enter.

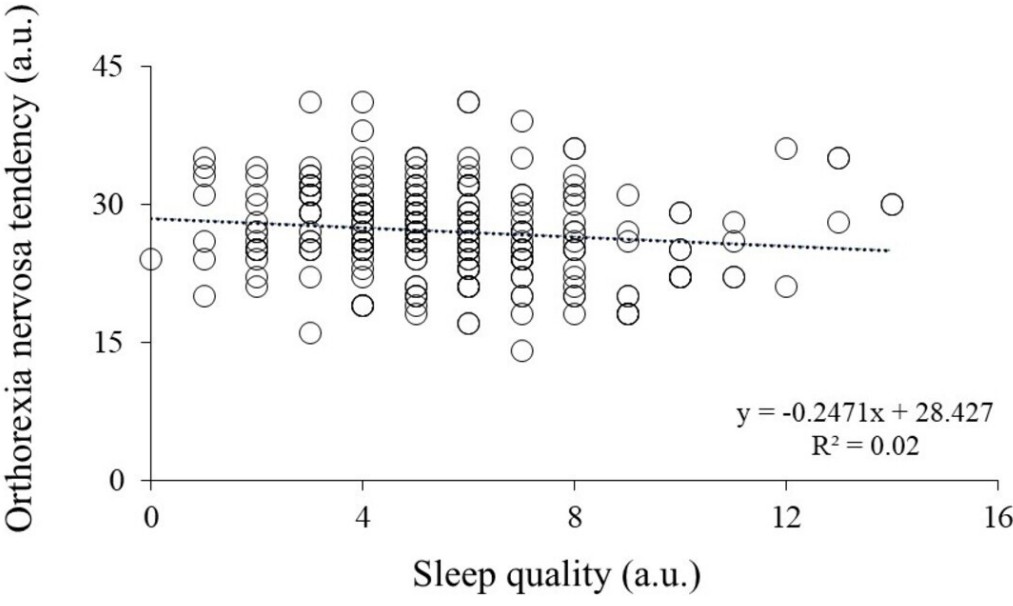

**Figure 1  Association between orthorexia nervosa tendencies (y-axis) and sleep quality (x-axis).** Higher scores indicate lower orthorexia nervosa tendencies and poorer sleep quality. Each point represents the score of an individual participant. a.u., arbitrary units.

**Table 2  Differences in orthorexia nervosa tendencies and sleep quality between individual and team athletes.** The second row in the table shows differences in sleep quality and the third row in orthorexia nervosa tendency between team and individual athletes.

| Variable | Branch | N | Median ± interquartile range | Mean rank | Sum of ranks | U | p |
|---|---|---|---|---|---|---|---|
| Sleep quality | *Individual* | 139 | 6 ± 3 | 139.45 | 19383.50 | 5636.500 | $\leq$.001* |
| | *Team* | 110 | 5 ± 3 | 106.54 | 11741.50 | | |
| Orthorexia Nervosa Tendency | *Individual* | 139 | 27 ± 7 | 120.68 | 16775.00 | 7045.000 | 0.287 |
| | *Team* | 110 | 27 ± 6 | 130.45 | 14350.00 | | |

Notes.

Higher median values indicating poorer sleep quality and a higher tendency of orthorexia nervosa.

*Indicate statistical significance.

findings revealed (1) that the athletes who present higher tendency of orthorexia nervosa are more likely to have poorer sleep quality (orthorexia nervosa tendency had significant negative predictive power on sleep quality explaining 2% of the mutual common variance),

(2) individual athletes were more likely to have poorer sleep quality compared to team athletes, however, no differences were observed in orthorexia nervosa tendencies between the two groups, and (3) athletes with longer sport tenure had poorer sleep quality. These results indicate that the elite athletes who present higher orthorexia nervosa tendency had higher risk for having poorer sleep quality, while the risk increases if they practice individual sport and have longer sport tenure.

Elite athletes presenting orthorexia nervosa tendency had a higher tendency of worse sleep quality. This finding is not surprising given that approximately 50% of individuals with eating disorders suffer from sleep disturbances (*Irish et al., 2024*). Although the mechanisms by which orthorexia nervosa affects sleep quality remain unclear, potential explanations may lie in the factors commonly associated with sleep disturbances in individuals with eating disorders. These include physical changes resulting from sudden weight loss, elevated levels of orexin leading to REM sleep disturbances, the presence of negative emotions and mood changes, body image concerns, etc. (*Degasperi et al., 2024*). It is important to acknowledge that the results of our study are based on a cross-sectional design, which precludes the establishment of a definitive causal relationship. Nonetheless, this topic warrants further investigation, as a clearer understanding of how orthorexia nervosa and eating disorders in general influence sleep quality could inform the development of more effective treatment approaches.

Individual and team athletes showed to present similar orthorexia nervosa tendencies. This is somewhat different to what has been reported in general when it comes to the differences in prevalence of eating disorders between individual and team athletes. Previous studies generally reported higher prevalence of eating disorders in individual compared to team athletes explaining that individual athletes are more prone to social judgments compared to team athletes (*Haase, 2009*). It has been also stated the successfulness in individual sports are more related to body aesthetics and weight compared to team sport athletes, and therefore perfectionistic behaviors in elite athletes make them prone to accepting extreme adherence to a certain diet. Some authors even point at perfectionism as a character trait to be responsible for orthorexia nervosa development (*Barnes & Caltabiano, 2017*; *Mavrandrea & Gonidakis, 2023*). When it comes to comparison between groups regarding sleeping quality, individual athletes were the one to obtain poorer scores. It is thought that the contradiction in the stated study findings is due to the difference in the sports branches included in the individual and team sports groups. In this context, it is recommended that future studies make inter-branch comparisons in addition to comparing team and individual sports.

There is no evidenced research for the orthorexia nervosa treatment. Some potential therapy includes cognitive-behavioral therapy (developing more balanced approach towards the food), exposure therapy (slowly reintroducing avoided food) and group therapy (sharing their struggles in a group) (*Douma, Valente & Syurina, 2021*). The treatment of orthorexia nervosa is complex and, as with other eating disorders, requires a multidisciplinary approach involving collaboration among family members, medical professionals, nutritionists, psychiatrists, and therapists. Coaches, due to their close and regular interaction with athletes, play a critical role in the early detection of disordered

eating behaviors. They are well-positioned to observe warning signs such as unexplained declines in performance or energy, significant weight loss, an excessive preoccupation with weight and appearance, obsessive checking of nutritional labels, and anxiety surrounding food. This role is particularly important for coaches of athletes in individual sports, where the risk of disordered eating behaviors tends to be elevated.

This study is not without limitations. First, it should be noted that some researchers have questioned the validity and reliability of the ORTO-15 scale, which was adapted into the ORTO-11 for the Turkish population (*Arusoğlu et al., 2008*). However, divided opinions regarding the validity of other scales used to assess orthorexia nervosa tendencies, such as the Düsseldorf Orthorexia Scale, Teruel Orthorexia Scale, and Bratman Orthorexia Test, complicate the selection of the most appropriate tool for accurate orthorexia nervosa diagnosis, as no gold standard exists (*Opitz et al., 2020*). Given our Turkish-speaking sample, we chose the validated Turkish version of the ORTO-11 as the most suitable. Nevertheless, it is widely acknowledged that a new scale is needed, one that addresses pathological preoccupation with healthy eating, emotional and psychological consequences of non-adherence to self-imposed dietary rules, and the associated risks of weight loss and malnutrition (*Cena et al., 2018*). Secondly, potential confounding variables such as socioeconomic status, health-related concerns, cultural differences, and pre-existing fitness or body image goals were not accounted for in this study. Thirdly, the cross-sectional design of the study limits the ability to establish causal relationships between orthorexia nervosa and sleep quality. Future research should focus on developing longitudinal studies to more accurately explore the directionality and underlying mechanisms of this association.

## CONCLUSIONS

The present study reveals that elite athletes presenting orthorexia nervosa tendency are prone to experiencing poorer sleep quality. This may be attributable to factors such as physical changes resulting from sudden weight loss, elevated orexin levels leading to REM sleep disturbances, negative emotional states, mood fluctuations, and concerns related to body image. Additional subgroup analysis revealed no significant difference in orthorexia nervosa tendencies between individual and team elite athletes, a finding that contrasts with earlier studies, which reported higher tendencies in individual athletes. However, individual athletes did exhibit poorer sleep quality compared to team athletes. Possible discrepancies might be explained by the great variability of sports included in the groups of individual (*i.e.,* tennis, wrestling, taekwondo and swimming) and team athletes (*i.e.,* football, basketball, volleyball, handball), while previous studies compared more homogenic groups. These results indicate that the elite athletes who present higher orthorexia nervosa tendency had a higher risk of having poorer sleep quality.

### Funding

This work was supported by a grant from the National Natural Science Foundation of China (grant number:12250410237). The funders had no role in study design, data collection and analysis, decision to publish, or preparation of the manuscript.

### Grant Disclosures

The following grant information was disclosed by the authors:
National Natural Science Foundation of China: 12250410237.

### Competing Interests

The authors declare there are no competing interests.

### Author Contributions

- Musab Çağın conceived and designed the experiments, performed the experiments, prepared figures and/or tables, and approved the final draft.
- Sezen Çimen Polat conceived and designed the experiments, analyzed the data, prepared figures and/or tables, and approved the final draft.
- Halil Sarol conceived and designed the experiments, performed the experiments, prepared figures and/or tables, and approved the final draft.
- Çisem Ünlü conceived and designed the experiments, analyzed the data, prepared figures and/or tables, and approved the final draft.
- Danica Janicijevic analyzed the data, prepared figures and/or tables, authored or reviewed drafts of the article, and approved the final draft.

### Human Ethics

The following information was supplied relating to ethical approvals (i.e., approving body and any reference numbers):

The Gazi University Ethics Commission granted ethical approval to carry out the present study (Code: 2023-1237)

### Data Availability

The data is available in the Supplemental File.

### Supplemental Information

Supplemental information for this article can be found online at http://dx.doi.org/10.7717/peerj.18349#supplemental-information.

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
