# Peer review of "A study on elite athletes: Orthorexia nervosa tendency is a risk factor for sleep quality"

_PeerJ, doi:10.7717/peerj.18349_

## Round 0.1 · original submission · Major Revisions

The study addresses a highly relevant issue in elite sports. While I appreciate the large sample size and the actuality of study there are strong concerns (specifically from Reviewer 2) regarding the suitability of the ORTO-11 scale. Please make sure to highlight these limitations and whether or not they may question the integrity of the results

·

Basic reporting

Clarity and Professionalism
• The paper is written in clear and professional English throughout. It is easy to understand and free from significant grammatical errors.
• Literature references are appropriate and provide sufficient background and context. However, more recent references could be included to enhance the relevance of the literature review.
Article Structure
• The article’s structure follows the standard format of scientific research papers, including an Abstract, Introduction, Methods, Results, Discussion, and Conclusion.
• Figures and tables are well-organized and appropriately labelled, aiding in the clarity of the results.
• The raw data is provided, enhancing the transparency and reproducibility of the study.
Self-Contained Nature
• The paper is self-contained and provides relevant results that address the stated hypotheses.
• The abstract summarizes the study well, including background, methods, results, and conclusions.

Experimental design

Originality and Scope
• The study presents original primary research that falls within the aims and scope of the journal.
• The research question is well-defined, relevant, and meaningful. The study addresses a gap in the literature regarding the relationship between orthorexia nervosa tendency and sleep quality among elite athletes.
Ethical Standards
• The investigation is conducted to a high technical and ethical standard. Ethical approval was obtained, and participants provided informed consent.
Methods
• The methods are described in sufficient detail to allow for replication. Validated scales (ORTO-11 and PSQI) should be used to measure orthorexia nervosa tendency and sleep quality, respectively.
• The statistical analyses are sound and appropriately chosen for the study’s aims.

Validity of the findings

Robust Data
• All underlying data are provided and appear robust and statistically sound. The results are well-supported by the data.
• The conclusions are well-stated and directly linked to the original research question and supporting results.
Impact and Novelty
• While the study's impact and novelty are not explicitly assessed, the findings provide meaningful contributions to the literature. The study encourages replication and further exploration of the relationship between orthorexia nervosa and sleep quality.

Additional comments

Comments and Recommendations
1. Literature Review: Consider including more recent studies in the literature review to ensure the context is up-to-date.
2. Discussion: Expand the discussion to include potential mechanisms linking orthorexia nervosa to sleep quality. Additionally, the implications of the findings for sports psychology and athlete management should be considered.
3. Limitations: Clearly state the study’s limitations. For example, acknowledge the potential influence of unmeasured confounding variables and the cross-sectional design's limitations in establishing causality.
4. Practical Implications: Discuss the practical implications of the findings for coaches, sports psychologists, and athletes. Providing recommendations for monitoring and managing orthorexia nervosa tendencies among athletes could enhance the paper's applied relevance.
5. Future Research: Highlight directions for future research, such as longitudinal studies to explore causality and interventions to mitigate the negative effects of orthorexia nervosa on sleep quality.
Comments on Table 1:
• The table is clear and well-organized, presenting the regression analysis results concisely.
• Including the model statistics (R, R², F, and p-values) is beneficial for understanding the strength and significance of the model.
• Explain the dependent variable and the method used to enhance the table's clarity.
• The table effectively shows that orthorexia nervosa tendency significantly predicts sleep quality, though the effect size (R² = 0.02) is relatively small.
Comments on Table 2:
• The table is well-structured and clearly shows the differences between individual and team athletes in terms of sleep quality and orthorexia nervosa tendency.
• The use of median and interquartile ranges is appropriate for these data types.
• Including mean ranks and the sum of ranks provides additional detail on the distribution of scores.
• The U and p-values are presented, indicating statistical significance where applicable.
• The explanatory note about higher median values and the significance indicator (*) helps interpret the results.
General Comments:
• Both tables align well with the paper's overall aim and help present the key findings concisely and clearly.
• The data presented in the tables support the conclusions drawn in the text, particularly regarding the impact of orthorexia nervosa on sleep quality and the differences observed between individual and team athletes.
• Minor formatting adjustments, such as aligning columns and ensuring consistent use of abbreviations (e.g., "f.001" should be "≤ 0.001"), would improve readability.
The tables are suitable for publication and effectively convey the necessary information. No major revisions are needed, though minor formatting improvements would enhance clarity.

Reviewer 2 ·

Basic reporting

no comment

Experimental design

In lines 85-93, the authors justify the purpose of their study and pose hypotheses.

"Although relatively frequent among elite athletes, to our knowledge there are no studies examining the relationship between sleep quality and orthorexia nervosa tendency in this population. "

The authors do not describe population differences in the introduction. Are they referring to differences in ON ? What does population have to do with the results presented in the study ?

The authors also hypothesize:

#1 high orthorexia nervosa tendency will negatively affect sleep quality

However, there is a lack of justification for this hypothesis - why do you bet that this should be a negative relationship ? Of what strength?

#2 while no hypothesis could be set regarding differences between athletes participating in individuals and group sports due to the lack of relevant studies.

Lack of studies is not a basis for no hypothesis - if there were already studies what would be the point ?

Please make a hypothesis and justify it (theoretically)

What about the seniority of the players? The authors analyze these results and did not make a hypothesis.

The ORTO-11 scale used mainly raises questions about this study. This scale and its properties are widely questioned. A meta-analysis found low overall reliability (α = 0.59) in 21 studies, suggesting the need for alternative assessment tools (Alshaibani et al., 2024). Another study identified issues related to the clarity and wording of items in the ORTO-15, noting discrepancies between participants' interpretations and ON's diagnostic criteria (Mitrofanova et al., 2020). Although it is still widely used, researchers caution against relying solely on the ORTO-15 in diagnosing ON and emphasize the need for a comprehensive assessment (Alshaibani et al., 2024; Mitrofanova et al., 2020).


Alshaibani, L., Elmasry, A., Kazerooni, A., Alsaeed, J.K., Alsendy, K., Alaamer, R.O., Buhassan, Z., Alaqaili, R., Ghazzawi, H.A., Pandi Perumal, S.R., Trabelsi, K., & Jahrami, H.A.. (2024). Reliability generalization meta-analysis of orthorexia nervosa using the ORTO-11/12/15/R scale in all populations and language versions. Journal of Eating Disorders, 12.

Mitrofanova, E., Pummell, E., Martinelli, L.A., & Petróczi, A. (2020). Does ORTO-15 produce valid data for 'Orthorexia Nervosa'? A mixed-method examination of participants' interpretations of the fifteen test items. Eating and Weight Disorders, 26, 897 - 909.

It is good practice to report the reliability of the scales/subscales used on the data you have collected. Reliability is a measure of the data collected. Please provide information on the obtained reliability of the scales used.

Did the collected data not allow the use of parametric tests?

With such small correlation levels, it would be good to include correlation graphs (possibly in the supplementary) so that we can be sure that the correlation is not due to an outlier, for example.

Is the correlation of sleep quality and ON symptoms different in a group of athletes involved in individual sports and group sports ?

Line 166-167 Why is statistical significance reported with a large P and not p as in other analyses


Line 150-153 The authors describe the methodology adopted to describe the strength of the correlation, but they do not use it, e.g.:
Line 163 rho= 0.131 described as "moderately" should be "small"

Validity of the findings

The discussion unfortunately needs to be completely revised. We cannot infer causality from correlation studies. The authors state causality from the very beginning (see line 174-175). The relationships obtained are so weak that even if it were an experiment it would be difficult to infer from them.

The discussion also lacks reference to the results in terms of seeking their justification/causation.

In my opinion, the conclusions presented in the discussion have no basis in the results presented.

Additional comments

no comment

---

## Round 0.2 · accepted · Accept

I am pleased to inform you that the manuscript now meets the requirements for acceptance and is ready for publication. The main concerns were satisfyingly addressed

·

Basic reporting

The authors replied to all comment, thank you

Experimental design

No comments

Validity of the findings

No comments